# Wide-field mid-infrared single-photon upconversion imaging

Kun Huang [1,2,3✉], Jianan Fang[1], Ming Yan[1,2], E Wu[1,2] & Heping Zeng [1,2,4,5✉]

Frequency upconversion technique, where the infrared signal is nonlinearly translated into the visible band to leverage the silicon sensors, offers a promising alternation for the mid-infrared (MIR) imaging. However, the intrinsic field of view (FOV) is typically limited by the phase-matching condition, thus imposing a remaining challenge to promote subsequent applications. Here, we demonstrate a wide-field upconversion imaging based on the aperiodic quasi-phase-matching configuration. The acceptance angle is significantly expanded to about 30°, over tenfold larger than that with the periodical poling crystal. The extended FOV is realized in one shot without the need of parameter scanning or post-processing. Consequently, a fast snapshot allows to facilitate high-speed imaging at a frame rate up to 216 kHz. Alternatively, single-photon imaging at room temperature is permitted due to the substantially suppressed background noise by the spectro-temporal filtering. Furthermore, we have implemented high-resolution time-of-flight 3D imaging based on the picosecond optical gating. These presented MIR imaging features with wide field, fast speed, and high sensitivity might stimulate immediate applications, such as non-destructive defect inspection, in-vivo biomedical examination, and high-speed volumetric tomography.

[1] State Key Laboratory of Precision Spectroscopy, East China Normal University, 200062 Shanghai, China. [2] Chongqing Key Laboratory of Precision Optics, Chongqing Institute of East China Normal University, Chongqing, China. [3] Collaborative Innovation Center of Extreme Optics, Shanxi University, 030006 Taiyuan, Shanxi, China. [4] Jinan Institute of Quantum Technology, 250101 Jinan, Shandong, China. [5] Shanghai Research Center for Quantum Sciences, 201315 Shanghai, China. ✉email: khuang@lps.ecnu.edu.cn; hpzeng@phy.ecnu.edu.cn

Mid-infrared (MIR) imaging is of particular scientific interest and technological importance in wide-ranging applications such as cosmic exploration, remote sensing, defect inspection, and medical diagnosis[1,2]. Typically, MIR cameras are available based on the narrow-bandgap semiconductors like HgCdTe and InSb[3]. However, these conventional imagers usually require the high-cost fabrication process based on epitaxial growth methods, as well as the stringent cryogenic operation[4]. In recent years, tremendous progress has been witnessed in developing sensitive MIR detectors at room temperature[5,6], especially resorting to emerging materials like colloidal quantum dots[7], black phosphorus[8,9], graphene[10], and tellurium nanosheet[11,12]. To date, the performance of the reported MIR photodetectors is still limited by the high dark current, which results in a noise equivalent power about nW/Hz$^{1/2}$, many orders of magnitude from the single-photon sensitivity[13]. Additional challenge lies in the improvement of the pixel number and the response time, aiming to approach the high-resolution MIR imaging at high-speed frame rates.

In contrast, silicon-based sensors in the visible band generally exhibit superior performances[14]. Particularly, single-photon detector arrays[15,16] and electron multiplying CCDs (EMCCDs)[17,18] enable to achieve ultra-sensitive imaging at the single-photon level. Recognizing the favorable features of Si-based cameras, the so-called frequency upconversion technique has attracted increasing attention as a simple yet effective alternation for the infrared imaging[19,20]. In this strategy, the intermediate optical conversion is typically realized by using a nonlinear crystal either based on the coherent frequency generation[21–23] or the quantum spectral correlation[24,25], where each spatial component within the object plane should satisfy the phase-matching condition to facilitate a pronounced conversion efficiency. In comparison to the direct imaging, the stringent phase-matching requirement would impose a constraint on the monochromatic field of view (FOV) of the upconversion imaging system, thus resulting in acceptance incident angles typically smaller than several degrees[20]. Moreover, the phase-matching restriction on the FOV becomes more pronounced at the presence of long crystals to enhance the nonlinear interaction length. The intrinsically small FOV of the upconversion imaging severely limit its promotion into broader applications.

So far, several methods have been proposed to enlarge the FOV of the upconversion imaging. One approach is to tune the phase-matching condition by varying the operating temperature of the nonlinear crystal, where all discrete phase-matched annular regions need to be stitched together to produce the full-FOV image[21,26,27]. The involved post-processing complexity is also manifested in the configuration involving the spatial translation of the object plane[28] or the angular rotation of the nonlinear crystal[29]. To circumvent the time-consuming procedure for the parameter variation, Maestre et al. proposed to implement a thermal gradient along the nonlinear crystal for broadening the angular acceptance of the upconverter[30]. In this case, the maximum FOV angle is intrinsically limited to about 10° according to the allowed phase-matching bandwidth at a feasible temperature difference. Another possible remedy can resort to the usage of broadband pump sources[31] or polychromatic signal illumination based on blackbody radiation[29], supercontinuum generation[32,33] and amplified spontaneous emission[34]. However, the non-constant wavelength scaling factor between the incident and upconverted fields would inevitably result in an imaging distortion with the radially dependent spatial magnification[20]. Very recently, Junaid et al. employed a high-speed Galvano scanner to rapidly dither the nonlinear crystal around the tangential phase-matching angle, which could lead to a 10-mm-diameter FOV within a 2.5 ms exposure time[35]. In this scenario, the frame rate is ultimately governed by the speed of the mechanical scanning since the camera

integration time is required to match the crystal rotation cycle time. Additionally, to support the sufficient rotation angle, a large transverse section of the bulky crystal is required, which may exclude the use of more efficient quasi-phase-matching (QPM) crystals with a typical thickness of 1 mm. We note that the MIR imaging has been directly recorded in silicon cameras based on the non-degenerate two-photon absorption[36]. The phase-matching-free operation could permit a large FOV, albeit with a low detection efficiency due to the intrinsically weak nonlinearity[37]. Therefore, it remains a long-standing challenge to access a large FOV while maintaining other desirable features for the upconversion imaging, particularly the single-photon sensitivity and high-speed frame rate.

Here we demonstrate a high-performance MIR upconversion imaging system with an unprecedented field of view up to about 30°. Our realization is based on a tailored QPM nonlinear crystal with a chirped-poling structure. The involved adiabatic conversion process enables to achieve a broad tolerance for the incidence angle by self-adapting the poling period. The resultant full FOV is realized in one acquisition at the unchanged settings, thus eliminating auxiliary procedures of the parameter tuning and the post-processing. Moreover, the combination of ultra-short pulse pumping and narrow-band signal illumination in the experiment favors to significantly suppress the background noise in the temporal and spectral domains. The ultimate sensitivity is verified by the single-photon imaging with an illumination intensity of 1 photon/pulse. In addition, an ultra-fast videography up to 216,000 frames per second is demonstrated to capture a moving target at a speed of about 31 m/s. Furthermore, the implemented MIR image upconverter is equipped with a picosecond coincident optical gating. This time-stamp functionality allows us to realize a high-resolution three-dimensional (3D) imaging by temporally selecting the reflected photons. Therefore, the presented wide-field upconversion imager is featured with high detection sensitivity, fast frame rate, and time-resolving capability, which would open up new possibilities in MIR applications.

## Results

**Basic principles**. The relevant mechanism for the upconversion imaging typically relies on the sum-frequency generation (SFG) in a nonlinear crystal. The involved fields satisfy the energy conservation law as $\hbar\omega_u = \hbar\omega_s + \hbar\omega_p$, where $\hbar$ is the Planck constant, $\omega_{s,p,u}$ are the optical angular frequencies for the signal, pump, and upconverted light. Additionally, the efficient frequency conversion requires the momentum conservation, i.e., nulling the phase mismatch $\Delta \vec{k} = \vec{k}_u - \vec{k}_s - \vec{k}_p$. To this end, the quasi phase matching (QPM) is commonly used to compensate the $\Delta \vec{k}$ by choosing an appropriate grating period $\Lambda$ for the periodically poled crystal[38]. The QPM technique favors a high conversion efficiency due to the possible access to a large nonlinear coefficient and an elongated interaction length in the collinear propagation[39,40]. Commonly, the periodically poled lithium niobate (PPLN) crystal has been employed to implement the upconversion imaging[21–23]. In this scenario, the imaging FOV is severely limited by the phase-matching requirement, as sketched in Fig. 1a. Figure 1b presents the theoretical simulation for a 10-mm-long crystal, which indicates a full acceptance angle of about 3.2°. Conceivably, the use of a longer crystal will impose a tighter confinement on the FOV due to the narrower phase-matching bandwidth[21,23,27,41]. In the simulation, the signal and pump wavelengths are set to be 3070 and 1030 nm, respectively. The crystal temperature is set to 51.7 °C for a period of 20.9 μm.

To address the limitation, a different configuration is proposed to significantly enhance the FOV of the upconversion imaging. The core element in this scheme is a chirped-poling lithium

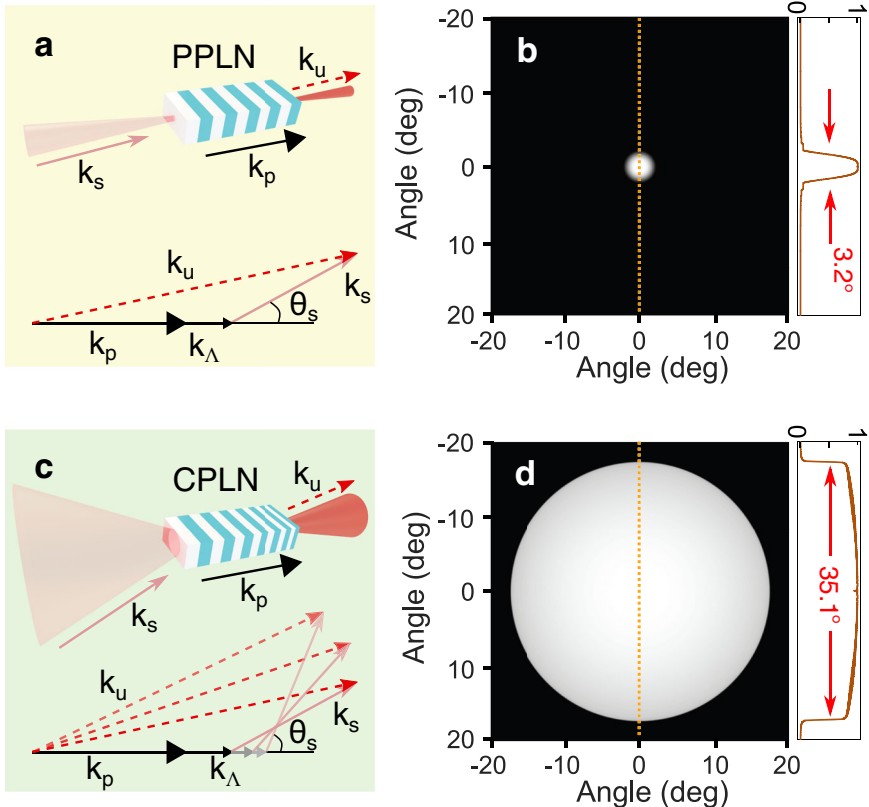

**Fig. 1 Principle and performance for the wide-field upconversion imaging. a** Conventional scheme for the upconversion imaging with a periodical-poling crystal, where the intrinsic field of view is severely limited by the phase-matching requirement. **b** presents the typical field of view of several degrees. The section profile corresponds to the dashed line in the image. **c** Proposed scheme to enlarge the field of view based on a chirped-poling nonlinear crystal. The tolerance for the incident angle is significantly improved due to the appropriate grating periods within a large adaptation range. **d** shows the theoretical performance. The full angle of the vision cone can be extended to 35.1°.

niobate (CPLN) crystal that is featured with a space-varied grating period along the propagation axis. This unique chirping structure has been used to implement the adiabatic nonlinear conversion with a large phase-matching bandwidth[42–44], thus facilitating parametric supercontinuum generation[45,46] and broadband upconversion detection[47,48]. Notably, the CPLN would provide higher efficiency compared to very short uniform gratings to achieve the same bandwidth[49]. A similar concept is adopted to extend the FOV as illustrated in Fig. 1c, where the phase mismatching for an oblique incidence can be effectively compensated by a proper periodicity for the nonlinearity grating. The underlying mechanism can be quantitatively investigated by decomposing the phase-mismatching wave vector into the transverse and longitudinal dimensions:

$$\frac{\Delta k_\parallel}{2\pi} = \frac{n_u}{\lambda_u}\cos\theta_u - \frac{n_s}{\lambda_s}\cos\theta_s - \frac{n_p}{\lambda_p} - \frac{1}{\Lambda}, \qquad (1)$$

$$\frac{\Delta k_\perp}{2\pi} = \frac{n_u}{\lambda_u}\sin\theta_u - \frac{n_s}{\lambda_s}\sin\theta_s. \qquad (2)$$

Generally, the refractive index $n$ for each optical field depends on the temperate $T$ and the angle $\theta$, which can be evaluated by the Sellmeier equation. For a given incident angle $\theta_s$ for the infrared signal, it is possible to solve the optimized angle $\theta_u$ for the upconverted beam and the corresponding poling period $\Lambda$ based on the phase-matching restriction of $\Delta k = \sqrt{\Delta k_\parallel^2 + \Delta k_\perp^2} = 0$. Specifically, according to Eq. (2) the incoming and outgoing angles should satisfy the following relationship at the perfect

phase-matching condition:

$$\frac{\theta_u}{\theta_s} \simeq \frac{\sin\theta_u}{\sin\theta_s} = \frac{\lambda_u}{\lambda_s} \times \frac{n_s}{n_u} \simeq \frac{\lambda_u}{\lambda_s}. \qquad (3)$$

This formula describes the angular magnification in the upconversion imaging, and can be used to infer the FOV at the object plane from the recorded upconverted image. Figure 1d shows the simulation result for a period range from 19 to 24 μm at the temperature of 30 °C. The resulting acceptance angle reaches to 35.1°, about tenfold larger than that based on the PPLN crystal.

We further characterize the wide-field upconversion imaging by varying the phase-matching parameters. Figure 2a presents the correspondence between the input external angle and the optimized poling period at different temperatures. It can be seen that the phase-matched ring in the object plane shrinks toward the center as increasing the temperature or the grating period. Additionally, the lower bound of the poling period $\Lambda_{\min}$ determines the largest allowed angle $\theta_s^{\max}$. Meanwhile, the upper bound $\Lambda_{\max}$ should exceed the phase-matched period for the normal incidence beam. In this case, all the rays within the angle $\theta_s^{\max}$ can be simultaneously converted to realize a full FOV. The horizontal dashed line in Fig. 2a denotes the limiting angle of the entrance pupil in the experimental imaging system. At the setting of $\Lambda_{\min} = 19$ μm, the full FOV angle is saturated to be 53° at the signal wavelength at 4.6 μm as given in Fig. 2b. This behavior is governed by the material dispersion of the nonlinear crystal[20]. In principle, a larger FOV can be obtained by using a smaller $\Lambda_{\min}$.

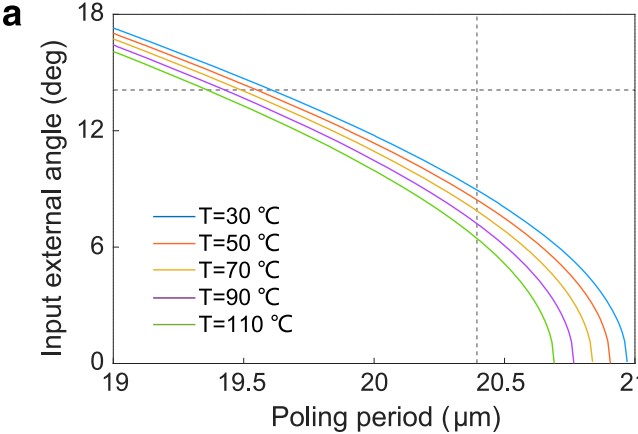

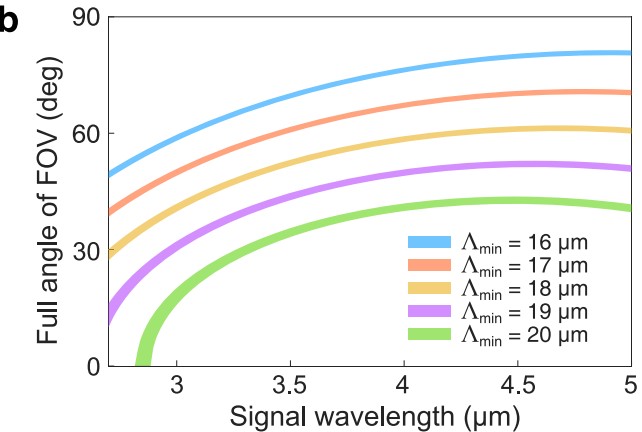

**Fig. 2 Dependence of the accepted angle on various phase-matching conditions. a** Input external angle as a function of the optimized poling period at different operation temperatures. The horizontal dashed line denotes the radius of the entrance pupil of the imaging system, while the vertical dashed line represents the condition for the upconversion imaging based on a PPLN crystal. **b** Full field-of-view angle versus the signal wavelength in the proposed wide-field scenario at the presence of various lower-bound poling periods. The vertical width for each plot band indicates the variation due to the temperature change from 30 to 110 °C.

**Experimental setup.** Figure 3 presents the artistic illustration of the experimental setup for the wide-field MIR upconversion imaging. The involved light sources originate from an ytterbium-doped fiber laser (YDFL) and an extended cavity diode laser (ECDL). The YDFL is mode-locked at the repetition rate of 21.6 MHz to prepare ultrashort pulses at 1030 nm. The average power can be boosted to 14 W after two stages of fiber amplifiers. The ECDL is a tunable narrow-band laser at the continuous-wave mode. The output power is increased to 100 mW by an erbium-doped fiber amplifier. Then, a portion of the YDFL output is spatially combined with the ECDL light by a dichroic mirror. The mixed beams are focused into a PPLN crystal to perform the difference frequency generation. The generated MIR pulse serves as the illumination source in the subsequent imaging. The 16-nm narrow spectrum of the MIR signal provides a much higher spectral brightness than the globar[29] or supercontinuum[32] sources. Additionally, the central wavelength can be tuned from 3015 to 3170 nm by continuously varying the emitting wavelength of the ECDL within the telecom C-band. In the temporal domain, the MIR pulse is self-synchronized with the pump source provided by the other portion from the YDFL. Therefore, the implemented dual-color light source offers a simple and compact

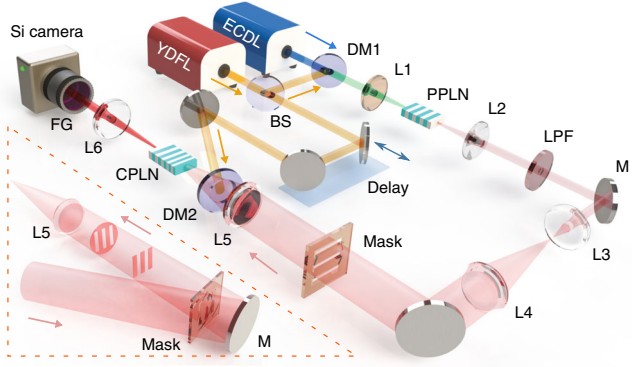

**Fig. 3 Experimental setup of the wide-field MIR frequency upconversion imaging.** The laser sources are from an ytterbium-doped fiber laser (YDFL) and a tunable extended cavity diode laser (ECDL) at the telecom C-band. The YDFL emits a train of mode-locked optical pulses at 1030 nm, while the ECDL operates at the continuous-wave mode. The dual-color beams are then injected into a periodically poled lithium niobate (PPLN) crystal to perform the difference-frequency generation for preparing the mid-infrared signal. The signal beam is then expanded with a pair of lenses before illuminating the object mask. Subsequently, the transmitted light is steered into a 4-f imaging configuration, where a chirped-poling niobate (CPLN) crystal is placed at the Fourier plane in order to realize the sum-frequency generation. Finally, the upconverted image after a spectral filter group (FG) is captured by a silicon camera. Notably, the temporal scanning of the synchronized pump pulse enables to realize the high-precision slicing of the volumetric imaging data, thus leading to the three-dimensional imaging. The required reflective imaging is configured based on the illumination fashion as shown in the bottom-left inset.

alternation to the optical parametric oscillator[35]. Details on the laser system are presented in Supplementary Note 1.

Subsequently, the synchronized signal and pump sources are used to implement the coincidence-pumping SFG. The MIR beam is enlarged by a beam expander before illuminating the object mask. The upconversion imaging system is configured in the 4f architecture, including two lenses with focal lengths of 50 and 100 mm. The formed object image is thus focused into a CPLN crystal of $10 \times 3 \times 1$ mm$^3$ at the Fourier plane. The 1-inch aperture of the lens results in a full acceptance angle of 28.5°. The pump beam has a diameter (full width at $1/e^2$) of 1.4 mm, larger than the crystal thickness. The cropped beam within the crystal shows a larger size along the horizontal direction than the vertical one, which leads to a slightly better horizontal resolution[50]. The Rayleigh length of the pump beam is much larger than the crystal length, which leads to a pump wave vector along the propagation axis. The CPLN is fabricated with a linearly-ramping poling period from 19 to 24 μm, which is designed to cover a broad spectral range from 2.7 to 5 μm (see Methods section). The chirped modulation of the QPM allows to realize the adiabatic nonlinear conversion, which has been proved to be robust to parameter variations of the nonlinear crystal and the incoming light[43]. Hence, the CPLN crystal can be operated at room temperature to avoid the high-precision oven as typically required for the PPLN crystal.

Then, the upconverted SFG signal is steered through a spectral filtering stage to remove the background noise from the pump-induced parametric fluorescence[23]. Finally, the filtered image is captured by a silicon-based camera. The zoom factor of the upconversion imaging system is about 0.5, which is used to restore the geometric dimension at the object plane. In the experiment, two types of imaging sensors based on the electron multiplying CCD (EMCCD) and the complementary metal-oxide

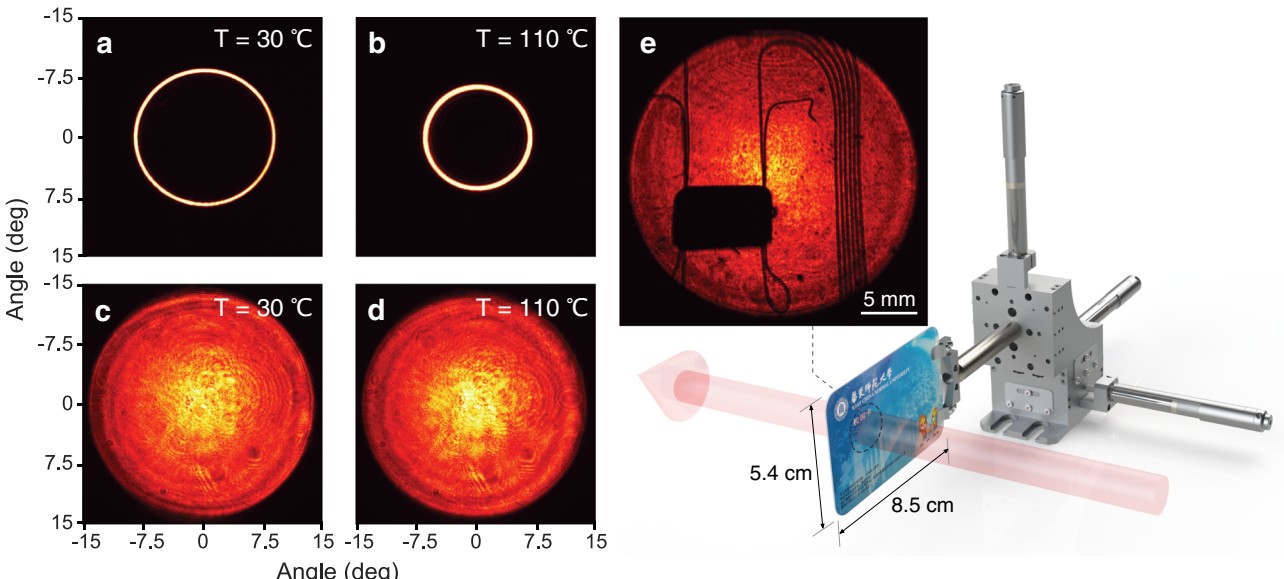

**Fig. 4 Wide-field MIR imaging performances. a**, **b** Recorded field of view for the conventional upconversion imaging with a PPLN at two different temperatures. **c**, **d** Wide-field performance for the imaging scheme with a CPLN crystal, which indicates a robust immunity to the temperature variation. **e** Transmissive imaging for a campus ID card. The embedded chip and mental wires can be clearly identified. A video is recorded in Supplementary Movie 1.

semiconductor (CMOS) are used to conduct a comprehensive characterization on the imaging system. Notably, besides the transmissive imaging, the illuminating modality is modified to implement the reflective imaging as shown in inset of Fig. 3. With the temporal scan of the pump pulse, a set of time-tagged imaging data can be acquired to reconstruct a 3D scenery.

**Wide-field MIR imaging.** Now we turn to characterize the upconversion imaging performance. Figure 4 presents the comparison on the available FOV with the PPLN and CPLN crystals. In the former case, annular regions are observed as shown in Fig. 4a, b, due to the limited acceptance bandwidth in the non-collinear phase matching[21,26]. The exhibiting ring would shrink toward the center as increasing the crystal temperature. This phenomenon can be understood from the vertical dashed line in Fig. 2a. Indeed, the input external angle decreases for a higher temperature at a fixed poling period. A full FOV can be approached with temperature tuning and image stitching[21]. Intrinsically, the allowed injection angle is limited to about 6° by varying the temperature from 30 to 110 °C, as identified in Fig. 2a at a period of 20.7 μm.

In contrast, the use of a CPLN crystal enables to achieve a wide FOV in one shot as shown in Fig. 4c, d, thus favoring a simpler and faster MIR image acquisition. Moreover, the FOV exhibits a strong resistance to the temperature change, which agrees well with the theoretical simulation in Fig. 2b (see also Supplementary Note 3). The visual scope at the object plane reaches to 24.8 mm, leading to an unprecedented full FOV angle of 28.2°. Note that the achieved acceptance angle is no longer limited by the nonlinear conversion. Instead, the entrance pupil of the imaging system and the transverse size of the crystal are the two main limiting factors to approach the ideal performance.

Figure 4e shows the proof-of-principle demonstration to reveal the interior structure of a campus ID card. The embedded chip and mental wires are identified through the polymer coverage (see Supplementary Movie 1). Pertinent to the transparency window for silicon and germanium materials, the MIR imaging would be useful in non-destructive defect inspection for semiconductor chips.

**Single-photon MIR imaging.** For the implemented image upconverter, the MIR laser illumination allows the use of narrow-band spectral filters to suppress the pump-induced noise. Besides, the coincident pulsed excitation provides an ultrashort time gate for the signal detection, thus significantly reducing the background noise from the ambient random scattering[51]. In order not to be limited by the dark noise of the camera itself, here we use a high-performance EMCCD to record the upconverted image. The MIR signal pulse is intensively attenuated to 40 photons/pulse before being injected into the upconversion imaging system. Figure 5a presents the image of a USAF-1951 resolution target with an integration time of 180 s. The spatial resolution is mainly determined by the crystal aperture, which filters out the high-frequency spatial components at the Fourier plane. The resolution is measured to be 125 μm as expected from the experimental settings. The achieved FOV of 24.8 mm in diameter thus leads to $3.1 \times 10^4$ resolvable spatial elements. Notably, the spatial resolution can be improved by magnifying the region of interest, albeit with a reduced FOV. As an example, a zoom factor of 4 would result in a resolution of 31 μm as shown in Fig. 5b. In addition, the pincushion distortion of the image is observed, which is ascribed to the 1-inch positive lens in the 4-f imaging system. The effect becomes noticeable when the beam size is close to the lens aperture. Figure 5d presents the corrected image for the lens distortion (see Methods section).

Under an illumination intensity of 1 photon/pulse, a longer exposure time of 600 s is used to acquire sufficient photons to form the image as shown in Fig. 5c. The background noise of the upconversion imaging system is mainly contributed by the pump-induced fluorescence, leading to a noise equivalent energy of about $2.2 \times 10^{-6}$ photons/pixel/pulse (see Supplementary Note 2). The achieved high sensitivity as well as the wide field and the high resolution would be desirable in applications like photon-starving infrared sensing, opaque material imaging, and phototoxicity-free biomedical examination.

**MIR imaging at ultra-high frame rates.** The wide-field imaging capability and low-noise conversion operation favor to obtain high-contrast images within a short exposure time at a modest illumination power, which can significantly improve the imaging

frame rate. In the experiment, a CMOS camera is used to validate the high-speed imaging competence. The object mask is replaced with an optical chopper rotating at a stabilized frequency of 100 Hz (see Supplementary Note 4). The radius of the rotational disk is about 5 cm, which results in a line speed of about 31.4 m/s for the far-end slot aperture. Figure 6a illustrates a sequential of

acquired images for the rotational chopper at the acquisition frame rate of 15 kHz. The exposure time is set to be 1.05 μs to minimize the motion-induced blurring effect for the fast-moving object. The illumination intensity is estimated to be about 5 mW/cm$^2$, which is much weaker than the previously reported results about mW/mm$^2$ [35]. The yellow arrow in Fig. 6a denotes the revolving direction and the updated position of the wheel. The frozen images at every 66.7 μs clearly show the temporal evolution of the chopper.

Furthermore, a faster frame rate at 216 kHz is used to characterize the chopper frequency up to 10 kHz. Figure 6b presents the recorded image. The frame size is decreased to $128 \times 80$ pixels, which is a common compromise to obtain a shorter time for the frame readout and the data manipulation in the camera. Figure 6c presents the time-dependent intensity on the denoted pixel in Fig. 6b. The period of 0.1 ms is consistent to the beam-switching cycle for the relevant category of slots. Note that the exposure time technically supports a frame rate up to MHz (see Methods section), which is about three orders of magnitude faster than commercially available MIR cameras. Such a high imaging speed would facilitate the time-resolved passive MIR imaging, pertinent to various researches in plasma formation, combustion process, fluid dynamics and so on. The recorded videos at various frame rates are compiled in Supplementary Movie 2.

**Three-dimensional MIR imaging.** The implemented MIR upconverter also enables to realize the time-of-flight imaging due to the coincident pulse pumping. Specifically, the picosecond pump pulse serves as an ultrafast optical gate of the upconversion imager, thus allowing to identify incoming photons with different time delays[52]. Consequently, the high-precision time-resolving property enables us to implement the 3D active imaging. In the proof-of-principle demonstration, the object is changed back to the 1951 USAF transmission resolution target. A silver mirror is placed behind the target as shown in inset of Fig. 3. Different from the previous setting for the transmission imaging, here we use the reflective illumination to form the stereoscopic object image. The negative resolution target is covered by the chrome coating, only leaving the patterns clear. The illumination beam is

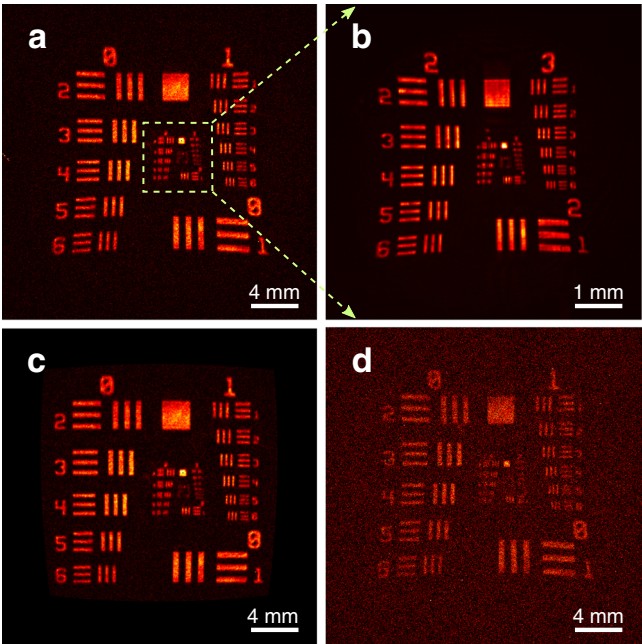

**Fig. 5 MIR single-photon imaging. a** MIR imaging for a resolution test target at the low-light-level illumination with 40 photons/pulse. The exposure time is set to be 180 s. **b** Enlarged object image with a zoom factor of 4, leading to an improved spatial resolution of about 31 μm. **c** Corrected image for the pincushion distortion. **d** Ultra-sensitive imaging at the single-photon operation with 1 photon/pulse. The exposure time is set to 600 s. Note that all the images are acquired by a highly-sensitive EMCCD.

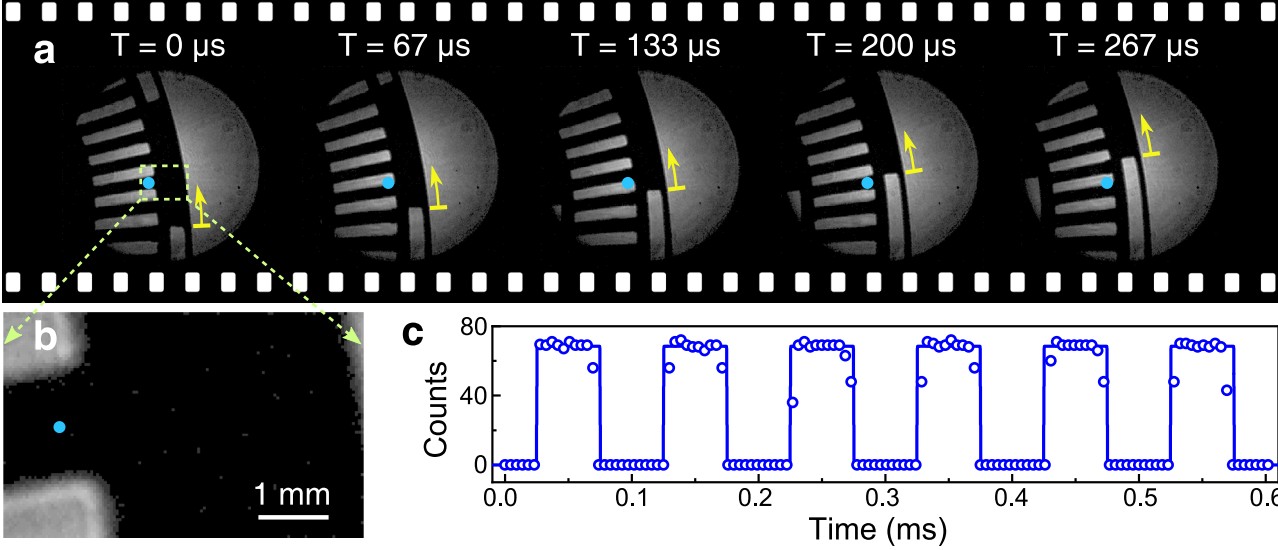

**Fig. 6 MIR imaging at high frame rates. a** Videography at 15,000 frames per second for a rapidly rotating chopper. The outermost slot rotates at a speed up to 31.4 m/s. **b** Video-shooting with a reduced frame size to support a faster frame rate up to 216,000 fps. **c** Temporal evolution of the intensity at the pixel denoted by a dot. The measured period of 0.1 ms is consistent with the 10-kHz angular frequency specified for the investigated slot. Note that all the images are acquired by a high-speed CMOS camera with an exposure time of 1.05 μs. The recorded videos are given in Supplementary Movie 2.

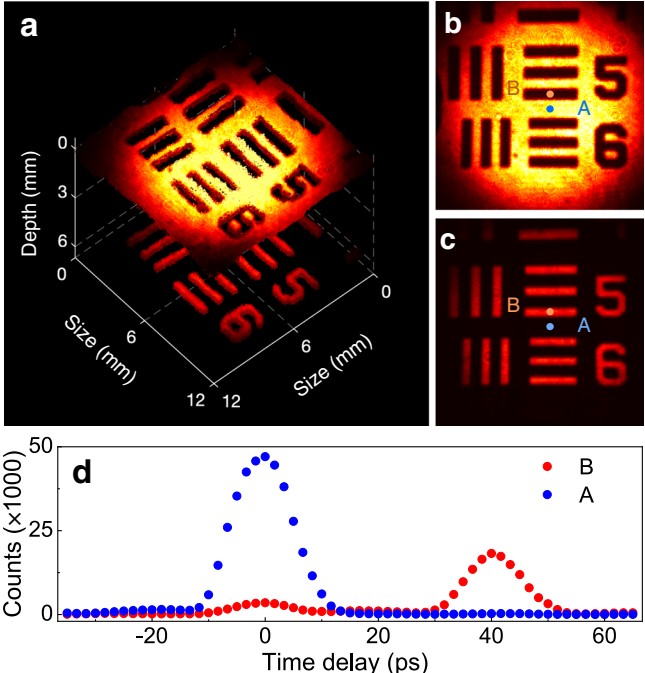

**Fig. 7 Proof-of-principle illustration of three-dimensional MIR upconversion imaging. a** Reconstructed 3D image for a resolution test target stacked on a silver mirror. The substrate of the negative target is coated with the chrome, only leaving the pattern itself clear. The transmitted light through the pattern is then reflected by the the silver mirror. The time-resolving ability of the implemented upconversion imaging system enables to differentiate the images from the front and back reflective surfaces, as shown in **b** and **c**, respectively. **d** presents the intensity evolutions for the two pixels labelled as A and B when varying the temporal delay of the pump gating pulse. The dynamic process is recorded in Supplementary Movie 3.

thus reflected from both the target substrate as the front surface and the silver mirror as the back surface. The temporally separated reflections are used to emulate the 3D scenery.

Figure 7 a presents the reconstructed 3D image from a set of volumetric data as varying the time delay of the pump pulse. The transition of the recorded images is manifested in Supplementary Movie 3. The longitudinal distance between the two surfaces is measured to be about 6 mm. As shown in Fig. 7b, the reflected image from the target substrate is inferred by maximizing the intensity at the pixel A on the recorded frame. Similarly, the mirror reflection given in Fig. 7c corresponds to the frame where the intensity of the pixel B is maximized. Furthermore, the intensity evolution for the pixels A and B are investigated as a function of the pump delay as shown in Fig. 7d. The time origin is set arbitrarily, and the scanning step is 1.7 ps. The two major peaks represent the refections from the two surfaces. The additional peak with a much smaller height is due to the residual reflection from the uncoated patterns of the resolution target. The full width at half maximum for the peaks is about 12 ps, which is determined by the duration of the cross-correlation between the signal and pump pulses (Supplementary Note 2). The achieved temporal resolution of the optical gating is much narrower than the available electronic gate, for instance the nanosecond gate in an intensified CCD. Moreover, the self synchronization between the signal and pump pulses results in a negligible relative timing jitter[51], which is essential in high-precision time-resolved analysis on the photoluminescence or other transient signals. Therefore, the implemented tomographic imaging provides an alternation to

the MIR optical coherence tomography for addressing an important need in the characterization of structured materials.

## Discussion

Frequency upconversion of infrared images can be traced back to the early years after the invention of the laser[19]. However, it has been only the last two decades that has witnessed the increasing attention payed to the upconversion technique[20]. Indeed, the upconversion imaging performance is greatly improved by the recent advances in high-efficiency QPM nonlinear crystals, high-power pump lasers, and high-sensitivity silicon cameras. To date, one of remaining challenges lies to enlarge the FOV of the upconversion imaging system, which is highly demanded to promote broader applications. Here we have addressed this long-standing quest to implement a wide-field MIR image upconverter based on the chirped QPM nonlinear crystal structure. In this configuration, the signal wave vectors at different angles can be simultaneously phase-matched with adapted poling periods. Equivalently, the extended acceptance angle allows to implement the MIR upconversion microscopy, where a high spatial resolution is possible by gathering more high-frequency components in the 2f imaging architecture.

Generally, the proposed method has three distinct features. First, the wide FOV is realized in one shot, without the need to change acquisition settings or post-process recorded images. Such a wide-field operation makes it possible to obtain a high-speed snapshot within a short exposure time of 1 µs. The resulting frame rate closed to MHz is much faster than the previous record at the kHz level limited by the millisecond integration time[35]. Second, the maximum FOV shows a weak dependence on the crystal temperature. In the laboratory environment, a nearly constant FOV has been observed, even though the nonlinear crystal is operated at room temperature. The simple and robust technique would promote the upconversion imaging into subsequent practical applications. Third, the large FOV is obtained with a narrow-band illumination, which favors to improve the overall conversion efficiency. In previous works with broadband sources[29,32,33], only the spectral component associated with a particular phase-matched angle can be efficiently upconverted. Also, the generated SFG signal has a narrow spectrum, which facilitates to use narrow spectral filters to suppress the background noise. The significantly improved signal-to-noise ratio is the key to achieve the sensitive imaging performance. Additionally, the use of a narrow-band signal eliminates the image distortion due to different wavelength scaling in the radial direction as appearing in the polychromatic illumination[32].

We note that the achieved FOV is technically limited by the entrance pupil of the 4f imaging system, where the illumination beam has been cropped by the 1-inch lens. Another potential restriction lies in the geometric dimensions of the used nonlinear crystal, which may exclude the large-angle injecting rays. The use of optical components with a larger aperture and shorter crystals with a wider section could in principle lead to a large FOV up to 60° at the presence of a chirping layout starting at 16 µm. Particularly, the use of thicker crystals allows us to employ a larger pump beam, which is favorable to improve the spatial resolution[20]. A high conversion efficiency is possible to be maintained by augmenting the pump power. In addition, the use of shorter pulses can further increase the pump intensity, and also improve the time resolution of the optical gating.

An immediate extension of the our imaging system is to implement the hyperspectral imaging in combination with a tunable MIR source (see Supplementary Note 5). Moreover, the proposed concept for the wide-field operation might be extended to the emerging

pattern-oriented QPM materials in the longer infrared wavelength regime[53]. We believe that the presented imaging modalities and performances would benefit MIR hyperspectral lifetime imaging with a high sensitivity, and pave a new way for high-throughput characterization of biological and material specimens[54].

## Methods

**Chirped-poling nonlinear crystal.** The CPLN crystal has a linearly increasing poling period from 19 to 24 μm along the length of 10 mm. The thickness and wideness are 1 and 3 mm, respectively. The crystal is antireflection-coated for the MIR signal wavelength ($R < 5\%$ @ 2.7–5 μm), pump wavelength ($R < 0.5\%$ @ 1.03 μm), and upconverted wavelength ($R < 0.5\%$ @ 745–854 nm). The crystal is operated at room temperature, thus resulting in a simple and robust operation. To go beyond the demonstrated performance, a larger section aperture for the crystal is possible to fabricate with the advanced poling technology, which favors to improve the entrance pupil and spatial resolution. In addition, a smaller starting period of the chirping range would help to obtain a larger FOV.

**4f imaging system.** The imaging system is based on the 4f configuration. The focal lengths of the two plano-convex lenses are 50 and 100 mm. The lenses are uncoated with the CaF$_2$ substrate, exhibiting an intrinsic transmission above 90% for a broad range from 0.35 to 7.6 μm. The actual zoom factor of the imaging system is inferred to be 0.5 by considering the angular magnification factor of about 0.25 due to the wavelength transduction. Limited by the aperture size of the entrance lens, the MIR illumination beam is cropped to a diameter of about 1 inch. The restricted FOV can be extended by using optical components with a larger size and nonlinear crystals with a wider section. The vertical and horizontal resolutions are measured to be 125 and 99 μm, consistent to the theoretical prediction. The spatial resolution can be improved by using an enlarged pump beam and a thicker crystal.

**Imaging deformation and correction.** The observed pincushion distortion in our experiment is ascribed to the 1-inch positive lens in the 4-f imaging system, instead of the intrinsic process for the nonlinear frequency conversion. Indeed, this phenomenon can also be observed for the imaging system without the conversion unit. The effect becomes more pronounced for a beam size closed to the lens aperture. Actually, such an imaging distortion is commonly seen for the cameras equipped with a wide-angle lens. Consequently, commercial photo-editing softwares, such as Adobe Photoshop and GIMP, already provide a handy tool to compensate the lens distortion. The corrected image is obtained by applying suitable algorithmic transformations to the digital photograph. To alleviate the pincushion distortion, one can use a lens with a larger aperture or resort to a special design of the surface curvature to compensate the radial-dependence magnification.

**Silicon-based cameras.** Two silicon-based cameras are used to characterize the upconversion imaging system. One is a back-illuminated EMCCD (Andor, iXon Ultra 888). The detection efficiency for the upconverted SFG signal at 771 nm is about 80%. With a spatial resolution of 13 μm, the full frame with $1024 \times 1024$ pixels permits a large field of view about 13.3 mm. The EMCCD is thermoelectrically cooled down to −80 °C to suppress the dark noise. The dark current of the EMCCD is specified to be about $10^{-4}$ electrons/pixel/second at a high gain of 1000. The superior sensitivity is favorable to demonstrate the MIR upconversion imaging at the single-photon level. To validate the high-speed videography, a CMOS-based camera (Photron, Mini AX200) is employed with a cutting-edge frame rate up to 900 kHz. Unfortunately, the performance is restricted to 216 kHz due to the export control regulations. The minimum exposure time is 1.05 μs, and can be improved to 260 ns in more advanced models. A 20-μm pixel pitch gives a sensor size of 20.48 mm at full image resolution. Limited by the speed fo readout electronics, the frame rate depends on the total number of involved pixels.

## Data availability

The data that support the findings of this study are available from the corresponding author upon reasonable request.

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

## Acknowledgements

This work was supported by the National Key Research and Development Program (2021YFB2801100), the National Natural Science Foundation of China (62175064, 11621404, 11727812), the Program for Professor of Special Appointment (Eastern Scholar) at Shanghai Institutions of Higher Learning, and the Fundamental Research Funds for the Central Universities.

## Author contributions

K.H. and H.Z. conceived the idea and designed the experiments. K.H. and J.F. built the system, performed experiments, and processed data. M.Y. built fiber laser sources. E.W. analyzed the imaging data. K.H. wrote the manuscript draft. All authors were involved in discussions and contributed to the manuscript editing.

## Competing interests

The authors declare no competing interests.
