## [Peer Review File · Nature Communications]

REVIEWER COMMENTS

Reviewer #1 (Remarks to the Author):

The authors report on upconversion imaging from the mid-infrared to the visible using a chirped-domain lithium niobate crystal (CPLN) and pump/illumination pulses in the picosecond regime.

There are several claims or points in this work. The main claim is to address a field of view (FOV) limitation typical of image upconversion systems by using a novel configuration based on a chirped-domain lithium niobate crystal. The manuscript reports a simulated 10-fold increase in FOV when compared with a PPLN crystal of the same length (1 cm), using monochromatic waves for modelling. In my opinion this is somehow not very useful in practice, not supported experimentally, and not novel as I will explain below.

The paper also provides insight into single-photon detection, fast frame rates, time of flight imaging through adjustment of a physical delay path, and adding temporal filtering to spectral filtering for improving S/N in single-photon detection.

The paper is well organized, clearly written and it is scientifically correct, although English should be checked and improved in my opinion. The experimental procedures are carefully realized and the supplementary images and videos provided are of good quality.

COMMENTS

It is known that image upconversion based on SFM (sum-frequency mixing) involves trade-offs among PM (phase matching) or QPM (quasi-phase matching) angle acceptance (FOV), conversion efficiency and resolution. While FOV is an important parameter in itself, the price paid in conversion efficiency for chirping makes a short length PPLN perform similar to CPLN in terms of FOV for the same efficiency. A short PPLN can provide essentially the same FOV for the same conversion efficiency, or alternatively, the same conversion efficiency for essentially the same FOV.

When comparing FOV in different situations, one must keep in mind that different theoretical/experimental situations lead to different absolute values. A real comparison in FOV

improvement techniques should be carried out keeping similar conditions (interacting wavelengths, their linewidths, pump peak power Vs. crystal length..). For instance, due to the lower derivative of the index dispersion at longer wavelengths, it is expected that angle acceptance (FOV) should be greater in the Mid-IR around 3 micron when compared to results obtained around 1550 nm for the same pump wavelength (say around 1040-1070 nm) in many cases. This follows clearly from the modelling in J. Warner, "Phase-Matching for Optical Up-Conversion with Maximum Angular Aperture-Theory and Practice", *Opto-Electronics* 1, 25-28 (1969). Type 0 QPM with a pump along a principal axis can be viewed as similar to a tangential phase matching.

REGARDING THE USE OF CPLN

The use of CPLN has been reported before in the context of image upconversion regarding the particular case of adiabatic sum-frequency mixing [43]. In Ref [42] a very large acceptance bandwidth is reported for CPLN. Although bandwidth acceptance is not the same as FOV, it is known in general to the community that a bandwidth increase in QPM leads also to a QPM acceptance angle (FOV) increase. Although for second harmonic generation, the basic principles about bandwidth and angle acceptances in QPM were reported in M.M. Fejer et al., "Quasi-phase-matched second harmonic generation: Tuning and tolerances", *IEEE J. Quantum. Electron.* QE-28, (1992) 2631–2654 (1992), where results derived for chirping are included particularly in relation with a strong efficiency reduction by chirping. This is not included in [38].

All in all, in H. Suchowski et al., "Robust adiabatic sum frequency conversion", *Opt. Express* 17(15), 12731-12740 (2009), not included in the manuscript, it is pointed out that QPM angle acceptance increases in CPLN. They checked theoretically that more than a 25 degree FOV can be achieved for SFM at a pump wavelength of 1064 nm and an IR signal (image) around 1550 nm. This value is presumably larger for a Mid-IR signal (image) around 3.1 micron due to the lower derivative of the index dispersion at longer wavelengths.

So in view of reference [43] and the one provided above by H. Suchowski, neither the use of a chirped PPLN crystal nor the increase in QPM acceptance angle (FOV) to the value reported in the manuscript in image upconversion can be viewed as a novel contribution or a breakthrough of this work.

REGARDING FOV

While the theoretical model of the authors predicts a 10-fold increase in FOV with respect to non-chirped PPLN for SFM (sum-frequency mixing) interaction between two monochromatic waves, the

experimental realization is made with a 1.8 nm linewidth pump wave and a 16 nm linewidth illumination wave. It is known that both non-monochromatic linewidths provide an increased FOV by themselves without the need of chirping. For instance, in [31], a pump wave 2.7 nm wide provides 110 mrad QPM of external acceptance angle (FOV) in a 2 cm long PPLN crystal whilst a narrowband pump < 0.1 nm provides a FOV of only 32 mrad in the same crystal, i.e., there is already a 4-fold increase in FOV by just increasing the pump linewidth up to 2.7 nm. In addition, if certain linewidth is used as well in the illumination source, the FOV is likely to increase even further according to [34]. In [31] the spectral width of the IR signal is not given (if I am not wrong), but because it is a CW fiber laser, it can be presumed that its linewidth will surely be much smaller than 16 nm. Because the authors use a 16 nm linewidth in the illumination source and 1.8 nm in the pump, the theoretically predicted 10-fold increase in FOV (from 3.2 degrees in PPLN to that experimentally obtained around 30 degrees) can not be attributed to chirping alone, and linewidths have an important contribution.

It is known that the QPM acceptance angle increases by shortening the length of the PPLN, so the equivalent acceptance in [34] extrapolated for a 10 mm long PPLN could be around 180-200 mrad. The FOV reported by the authors is 520 mrad, so it is not so remarkably (roughly only 2-3-fold) above that achievable in PPLN even for the same crystal length under similar conditions, and far from the theoretical 10-fold achievable.

However, because chirping the domain size in CPPLN leads to a reduction in the nonlinear effective coefficient or alternatively conversion efficiency with regard to PPLN (not discussed in detail in the text), a comparison with a short length PPLN crystal can be made. According to the 6 W average power, the 10 ps pulse duration, and the pulse repetition frequency of 26.1 MHz, the peak power in each pulse must be around 28 kW, leading to a conversion efficiency of 0.1 % as described in the Supplementary Material file. Using the standard plane-wave model for SFM efficiency [*], this peak power for a 1 mm diameter beam would produce the same 0.1 % conversion efficiency in a PPLN crystal of length ≈ 0.1 mm. If the authors compute the FOV in a pure PPLN crystal 10 mm long (3.2 degrees, given in the text) and compare it with a PPLN crystal 0.1 mm long that leads to the same conversion efficiency, they will find out essentially the same 10-fold FOV enhancement even for monochromatic waves. For this reason the way of achieving the 30 degrees FOV reported seems to be not so useful in practice.

[*] The well-known Boyd-Kleinman efficiency expression for interactions among optimally focussed gaussian beams given in G.D. Boyd, D.A. Kleinman, "Parametric Interaction of Focused Gaussian Light Beams", J. Appl. Phys. 39, 3597 (1968), can not be directly applied due to the pump beam diameter used in relation to the length of the crystal.

Regarding QPM acceptance angle and efficiency with chirping Vs. short PPLN, some analysis can be found in A. Bostani et al., "Super-tunable, broadband upconversion of a high-power CW laser in an engineered nonlinear crystal", Sci. Rep. 7, 883 (2017). There, a full acceptance angle of 23 degrees

for a 13 mm long chirped crystal with an IR wavelength around 1550 nm (presumably greater for 3.1 micron, and a shorter 10 mm crystal).

REGARDING SINGLE-PHOTON

Mid-IR image upconversion at the single photon level is not a breakthrough either. It was already demonstrated in [21]. However, it was done in CW using only spectral filtering. Because the system is operated in pulsed mode, an additional temporal filtering is added in this work which improves S/N. However, this idea of enhancing S/N by pulse synchronization was previously reported by some of the present co-authors in [18] regarding few-photon image upconversion detection, so it can not be viewed as a new contribution of impact.

REGARDING TIME-OF-FLIGHT IMAGING BY PULSE COINCIDENCE

As mentioned above, some of the co-authors of this work already presented the idea of pulse coincidence for few photon detection in [18], which directly adds time-of-flight imaging capabilities. Although it is expected that the need of pulse coincidence (also used in [*]) adds time-of-flight imaging capabilities, the authors present a nice example in a supplementary video by making a transmission and reflection IR image of a target image enter the upconversion system and using time delay discrimination based on a physical change of paths.

SCIENTIFIC QUALITY

The manuscript is scientifically correct in my opinion, although there are some minor issues that I comment next:

The 1.4 mm diameter beam is cropped to 1 mm only in the vertical direction. This flattens the intensity distribution of the pump beam in that direction (the flattening is not too strong, so the term flat-top may be questionable). The horizontal direction becomes uncropped, and has a slightly larger (1.4 times) beam diameter. It is only in the horizontal direction that resolution can be

increased due to the larger soft gaussian aperture in the Fourier plane. In my opinion this should be commented in the text.

Assuming an uncropped 1 mm pump beam diameter at $1/e^2$ intensity and a 1.4 mm beam cropped to 1 mm in the vertical direction, there is not a difference in the highest spatial frequency allowed through the system. Only relative amplitudes of the plane wave Fourier components change (slightly). This may represent only a slight difference (raise) in contrast for intermediate spatial frequencies only. This is discussed in Chen Yang et al., "Frequency up-conversion of an infrared image via a flat-top pump beam", Opt. Comm. 460, 125143 (2020).

Simply stating in the text that "The cropped beam leads to a flat-top profile, thus improving the spatial resolution" is not strictly correct and requires clarification and/or referencing.

Other issues:

If I am not wrong, the focal length F_2 is not given in the text or in the additional material.

In fig. 3 (k to o) it is shown that the CPLN is robust against temperature variation. This is an experimental verification of what is already outlined in ref [42]. The information in (a to j) is quite similar to that given experimentally and theoretically in [30].

SUMMARY

For the reasons I give, rather than a new contribution of impact, this work is in essence the combination of methods used in previous works by other authors or by some co-authors of this work. Therefore, this paper does not represent in my opinion an important advance or breakthrough to specialists in the field of significance enough for the usual standards of Nature Communications. There are some welcome experimental performance improvements in this work however, but only of an incremental character for specialists in the field.

It is worth publishing the article in another journal, as image up-conversion is a valuable rescued field from the late sixties and all information, progress, or reviews in the field, are valuable in my opinion.

Reviewer #2 (Remarks to the Author):

Summary:

The authors present a paper for increasing the angular bandwidth of image upconversion when using monochromatic illumination; A paper which I read with great interest. Using a chirped PPLN structure (CPLN), the authors demonstrate that a large field of view (FOV) can be obtained (35.1 degrees) versus 3.2 degrees for a standard PPLN crystal (10 mm long). This is obtained without any moving parts contrasting, for instant, reference [35]. The large FOV is a highly attractive feature for many MIR applications and the demonstrated FOV approaches that of standard cameras which is excellent. The FOV is the main scientific accomplishment and provides a significant advancement compared to prior art. A special feature is the high temperature stability, which promotes easy implementation of the crystal assembly. Secondly, the authors demonstrate central features of their imaging approach for different applications. a) Impressive image acquisition update rate of 214 KHz b) Demonstration of single photon sensitivity and c) 3-D imaging. These demonstrations adds significant value to the work serving as inspiration for new implementations and applications (although not entirely new).

Technical comments:

Upconversion efficiency: The “price” for obtaining a large FOV using CPLN in the presented approach is a reduction in upconversion conversion efficiency compared to co-linear monochromatic upconversion using a PPLN with the same crystal length. A back of the envelope calculation suggests that the reduction in efficiency may well be in the order of 100 $(35.1/3.2)^2$. This is tolerable for some implementations, however important to consider for other situations. If possible,

- I suggest that the authors include a few remarks about the efficiency of using their CPLN implementation, preferably quantitative.

Set-up: The set-up includes a mode-locked laser, which mixes (DFG) with a CW ECDL to generate the MIR signal. The DFG peak efficiency, I expect, will be high due to the high peak power present in the YDFL, however the small duty cycle of the mode-locked laser will diminish average conversion efficiency to $< < 1\%$. This can be overcome by synchronous pulsed upconversion where pump pulse and mixing pulse is temporally matched, e.g. [35]

- The authors should list the pulse width of the YDFL pulses and preferably provide a few comments on the efficiency.

The authors nicely demonstrate the possibility to temporal filter noise using the short pulse as a time window for noise elimination. The idea is not new, but illustrate the single photon capability.

Aberrations: Using a chirped CPLN to upconvert with a large FOV imply that different angles will be upconverted in different sections of the CPLN crystal. This is like a “thick” filter.

- Will this give rise to image aberrations? The authors should include a few comments on this issue.

Conclusion: The paper is clearly written and easy to understand, figures and graphs are of high quality. When considering the paper in its totality, it merits publication in Nature Comm. after minor revision of different points as listed above.

Manuscript NCOMMS-21-34014
“Wide-field mid-infrared single-photon upconversion imaging”
Reply to the Reviewers

We would like to thank the two reviewers for the careful reading of the manuscript and their valuable reports. We give below a detailed response to the reviewers’ comments. Excerpts from the original reports are given in blue. Changes in the revised manuscript are indicated in green.

Reviewer #1

The authors report on upconversion imaging from the mid-infrared to the visible using a chirped-domain lithium niobate crystal (CPLN) and pump/illumination pulses in the picosecond regime.

There are several claims or points in this work. The main claim is to address a field of view (FOV) limitation typical of image upconversion systems by using a novel configuration based on a chirped-domain lithium niobate crystal. The manuscript reports a simulated 10-fold increase in FOV when compared with a PPLN crystal of the same length (1 cm), using monochromatic waves for modelling. In my opinion this is somehow not very useful in practice, not supported experimentally, and not novel as I will explain below.

We thank the reviewer for his/her careful reading of the manuscript and giving us these valuable comments. Our work addresses the long-standing quest for the infrared upconversion imaging to realize a large field of view while maintaining a high imaging sensitivity. The single-shot FOV reaches to an unprecedented value about 30° , which even twice larger than the previous record obtained with a mechanical scanner [R1]. The substantially improved FOV in comparison to the conventional PPLN has been verified both theoretically and experimentally. Additionally, the combination with the tight spectro-temporal filtering enables us to achieve the single-photon capability. The highly-sensitive feature facilitates the high-speed imaging at a frame rate up to 216 kHz, which is orders of magnitude faster than any reported MIR imaging systems. These demonstrated state-of-the-art features are made possible by the novel configuration and original implementation, which represent significant and firm advances in the field of MIR imaging. Notably, the CPLN-based upconversion imager intrinsically supports the broadband operation to go beyond the monochromatic demonstration, as already demonstrated preliminarily for a tunable range from 3015 to 3170 nm in Supplementary Note 5. Therefore, the presented work could immediately favor subsequent applications like non-destructive defect inspection, in-vivo biomedical examination, and high-speed volumetric tomography.

[R1] Junaid, S., Chaitanya Kumar, S., Mathez, M., Hermes, M., Stone, N., Shepherd, N., Ebrahim-Zadeh, M., Tidemand-Lichtenberg, P. & Pedersen, C. Video-rate, mid-infrared hyperspectral upconversion imaging. *Optica* 6, 702-708 (2019).

The paper also provides insight into single-photon detection, fast frame rates, time of flight imaging through adjustment of a physical delay path, and adding temporal filtering to spectral

filtering for improving S/N in single-photon detection.

The paper is well organized, clearly written and it is scientifically correct, although English should be checked and improved in my opinion. The experimental procedures are carefully realized and the supplementary images and videos provided are of good quality.

We thank the reviewer for recognizing the achieved superior MIR imaging features of wide field, fast speed, and high sensitivity in this work. These achievements are considered to be well supported by the solid data and clear plots.

In the following, we give answers to the points raised by the reviewer and describe the changes we made in the manuscript to take into account these valuable comments, which helped us to improve further the clarity of the paper. The writing has also been polished.

COMMENTS

It is known that image upconversion based on SFM (sum-frequency mixing) involves trade-offs among PM (phase matching) or QPM (quasi-phase matching) angle acceptance (FOV), conversion efficiency and resolution. While FOV is an important parameter in itself, the price paid in conversion efficiency for chirping makes a short length PPLN perform similar to CPLN in terms of FOV for the same efficiency. A short PPLN can provide essentially the same FOV for the same conversion efficiency, or alternatively, the same conversion efficiency for essentially the same FOV.

We agree with the reviewer that a shorter PPLN can offer a larger phase-matching bandwidth. However, the conversion efficiency decreases quadratically with the reduction of the crystal length. Consequently, the broadband operation usually leads to prohibitively low conversion efficiencies. It is this notorious constraint that renders the long-standing challenge to realize a upconversion imager with a large FOV and a high sensitivity. In contrast, the CPLN favors the adiabatic nonlinear conversion, which could thus *“provide higher efficiency compared to very short uniform gratings to achieve the same bandwidth”* as stated in Ref. R2. In our work, the spirit of using the chirping structure has for the first time been adopted to facilitate a wide-field operation for the infrared image upconversion.

In the revised manuscript, this advantage is now indicated in the text: *“This unique chirping structure has been used to implement the adiabatic nonlinear conversion with a large phase-matching bandwidth, thus facilitating parametric supercontinuum generation and broadband upconversion detection. Notably, the CPLN would provide higher efficiency compared to very short uniform gratings to achieve the same bandwidth [R2].”*

[R2] Bostani, A., Tehrani, A. & Kashyap, R. Super-tunable, broadband up-conversion of a high-power CW laser in an engineered nonlinear crystal. *Sci. Rep.* 7, 883 (2017).

When comparing FOV in different situations, one must keep in mind that different theoretical/experimental situations lead to different absolute values. A real comparison in FOV improvement techniques should be carried out keeping similar conditions (interacting

wavelengths, their linewidths, pump peak power Vs. crystal length..). For instance, due to the lower derivative of the index dispersion at longer wavelengths, it is expected that angle acceptance (FOV) should be greater in the Mid-IR around 3 micron when compared to results obtained around 1550 nm for the same pump wavelength (say around 1040-1070 nm) in many cases. This follows clearly from the modelling in J. Warner, “Phase-Matching for Optical Up-Conversion with Maximum Angular Aperture-Theory and Practice”, *Opto-Electronics* 1, 25-28 (1969). Type 0 QPM with a pump along a principal axis can be viewed as similar to a tangential phase matching.

Indeed, the FOV based on periodical poling crystals would be larger for longer signal wavelengths, but still limited within small angles. For instance, the achieved FOV at 1.55 μm and 3.1 μm are about 0.6° [R3] and 3° [R4], respectively. The achieved single-shot FOV up to 30° in this manuscript thus presents substantial improvement among all the reported protocols at similar operation wavelengths.

[R3] Liu, S., Yang, C., Liu, S., Zhou, Z., Li, Y., Li, Y. H., Xu, Z. H., Guo, G. & Shi, B. Up-conversion imaging processing with field-of-view and edge enhancement. *Phys. Rev. Appl.* 11, 044013 (2019).

[R4] Dam, J. S., Tidemand-Lichtenberg, P. & Pedersen, C. Room-temperature mid-infrared single-photon spectral imaging. *Nat. Photon.* 6, 788-793 (2012).

REGARDING THE USE OF CPLN

The use of CPLN has been reported before in the context of image upconversion regarding the particular case of adiabatic sum-frequency mixing [43]. In Ref [42] a very large acceptance bandwidth is reported for CPLN. Although bandwidth acceptance is not the same as FOV, it is known in general to the community that a bandwidth increase in QPM leads also to a QPM acceptance angle (FOV) increase. Although for second harmonic generation, the basic principles about bandwidth and angle acceptances in QPM were reported in M.M. Fejer et al., “Quasi-phase-matched second harmonic generation: Tuning and tolerances”, *IEEE J. Quantum. Electron.* QE-28, (1992) 2631–2654 (1992), where results derived for chirping are included particularly in relation with a strong efficiency reduction by chirping. This is not included in [38].

Indeed, our implemented configuration is inspired by the concept of adiabatic conversion based on a chirped-poling crystal, as already stated in the manuscript. In previous works, the discussion on the acceptant bandwidth is related to the broadband spectral conversion. To the best of knowledge, there is no theoretical proposal or experimental demonstration in applying this technique to achieve a large-FOV upconversion imaging. For example, Ref. [43] only focuses on the multicolor operation of the upconversion imager, without mentioning the FOV or imaging sensitivity of their system. Our work here provides a simple yet effective solution to dramatically boost the performances of the upconversion imaging system.

All in all, in H. Suchowski et al., “Robust adiabatic sum frequency conversion”, *Opt. Express* 17(15), 12731-12740 (2009), not included in the manuscript, it is pointed out that QPM angle acceptance increases in CPLN. They checked theoretically that more than a 25 degree FOV can

be achieved for SFM at a pump wavelength of 1064 nm and an IR signal (image) around 1550 nm. This value is presumably larger for a Mid-IR signal (image) around 3.1 micron due to the lower derivative of the index dispersion at longer wavelengths.

Again, the mentioned work merely focuses on the broadband spectrum conversion without showing the imaging performance. For the first time, we theoretically and experimentally demonstrate that a large-FOV imaging can be obtained with a narrow-band illumination. In this context, the scope of our work is totally different from the aforementioned report. For completeness, the reference has been added in the revised manuscript: *“This unique chirping structure has been used to implement the adiabatic nonlinear conversion with a large phase-matching bandwidth [R5], ...”*

[R5] Suchowski, H., Prabhudesai, V., Oron, D., Arie, A. & Silberberg, Y. Robust adiabatic sum frequency conversion. Opt. Express 17, 12731-12740 (2009).

So in view of reference [43] and the one provided above by H. Suchowski, neither the use of a chirped PPLN crystal nor the increase in QPM acceptance angle (FOV) to the value reported in the manuscript in image upconversion can be viewed as a novel contribution or a breakthrough of this work.

Both works as pointed by the reviewer focused on demonstrating the frequency upconversion with a broad spectral bandwidth. However, the scope of our work concentrates the imaging performance in the spatial domain, which is conceptually different from previous demonstrations. In combination with noise filtering techniques, we demonstrate unprecedented MIR imaging performance with wide field, fast speed, and high sensitivity, which is considered as a breakthrough of this work.

REGARDING FOV

While the theoretical model of the authors predicts a 10-fold increase in FOV with respect to non-chirped PPLN for SFM (sum-frequency mixing) interaction between two monochromatic waves, the experimental realization is made with a 1.8 nm linewidth pump wave and a 16 nm linewidth illumination wave. It is known that both non-monochromatic linewidths provide an increased FOV by themselves without the need of chirping. For instance, in [31], a pump wave 2.7 nm wide provides 110 mrad QPM of external acceptance angle (FOV) in a 2 cm long PPLN crystal whilst a narrowband pump < 0.1 nm provides a FOV of only 32 mrad in the same crystal, i.e., there is already a 4-fold increase in FOV by just increasing the pump linewidth up to 2.7 nm. In addition, if certain linewidth is used as well in the illumination source, the FOV is likely to increase even further according to [34]. In [31] the spectral width of the IR signal is not given (if I am not wrong), but because it is a CW fiber laser, it can be presumed that its linewidth will surely be much smaller than 16 nm. Because the authors use a 16 nm linewidth in the illumination source and 1.8 nm in the pump, the theoretically predicted 10-fold increase in FOV (from 3.2 degrees in PPLN to that experimentally obtained around 30 degrees) can not be attributed to chirping alone, and linewidths have an important contribution.

We agree with the reviewer that the FOV can be enlarged by using broadband pump sources or

polychromatic signal illumination, as discussed in the introductory part. However, the improvement is very limited. Particularly, under the 1030-nm pump, the FOV of 30° requires the a signal coverage up to about 700 nm [c.f. J. Dam *et al.*, Nature Photon. 6, 788 (2012)]. To verify the contribution due to the laser parameters, we experimentally measured the FOV for the PPLN and CPLN crystals as shown in Figure R1. The measured results agree well with the theoretical simulation, thus validating that the chirping structure indeed plays a dominant role in the FOV enlargement.

In the revised version, we add the figure and related discussion in Supplementary Note 2.

Figure R1: Recorded field of view for the PPLN (a) and CPLN (b) crystals under the same signal and pump settings. Both crystals have a length of 10 mm.

It is known that the QPM acceptance angle increases by shortening the length of the PPLN, so the equivalent acceptance in [34] extrapolated for a 10 mm long PPLN could be around 180-200 mrad. The FOV reported by the authors is 520 mrad, so it is not so remarkably (roughly only 2-3-fold) above that achievable in PPLN even for the same crystal length under similar conditions, and far from the theoretical 10-fold achievable.

In Ref. [34], the FOV in the collinear interaction is limited to 1.6° (~ 30 mrad) for a crystal length of 5 mm. The FOV under the ASE illumination is shown to be slightly larger, but the specific value is not given in the text. According to Fig. 5 in that paper, the improvement for a broadband signal is less than 1.5 times, leading to a FOV smaller than 45 mrad. For a longer crystal length of 10 mm, the extrapolated FOV should be reduced.

The achieved FOV up to 30° (~ 520 mrad) is significantly improved comparing to the previous reports. Additionally, the imaging sensitivity and frame rate in our work is orders of magnitude better.

However, because chirping the domain size in CPPLN leads to a reduction in the nonlinear effective coefficient or alternatively conversion efficiency with regard to PPLN (not discussed in detail in the text), a comparison with a short length PPLN crystal can be made. According to the 6 W average power, the 10 ps pulse duration, and the pulse repetition frequency of 26.1 MHz, the peak power in each pulse must be around 28 kW, leading to a conversion efficiency of 0.1 %

as described in the Supplementary Material file. Using the standard plane-wave model for SFM efficiency [*], this peak power for a 1 mm diameter beam would produce the same 0.1 % conversion efficiency in a PPLN crystal of length ~ 0.1 mm. If the authors compute the FOV in a pure PPLN crystal 10 mm long (3.2 degrees, given in the text) and compare it with a PPLN crystal 0.1 mm long that leads to the same conversion efficiency, they will find out essentially the same 10-fold FOV enhancement even for monochromatic waves. For this reason the way of achieving the 30 degrees FOV reported seems to be not so useful in practice.

[*] The well-known Boyd-Kleinman efficiency expression for interactions among optimally focussed gaussian beams given in G.D. Boyd, D.A. Kleinman, “Parametric Interaction of Focused Gaussian Light Beams”, J. Appl. Phys.39, 3597 (1968), can not be directly applied due to the pump beam diameter used in relation to the length of the crystal.

The reviewer mistook the pump power. Actually, the conversion efficiency of 0.1% is obtained at 0.6-W pump power, as described in Supplementary Note 2. According to the reviewer’s calculation, the use of 0.1-mm-length PPLN crystal will lead to much lower efficiency at the condition of the same bandwidth. Consequently, our proposed imaging configuration based on CPLN is indeed advantageous to obtain a higher efficiency over the scheme simply using a thin PPLN crystal.

Regarding QPM acceptance angle and efficiency with chirping Vs. short PPLN, some analysis can be found in A. Bostani et al., “Super-tunable, broadband upconversion of a high-power CW laser in an engineered nonlinear crystal”, Sci. Rep. 7, 883 (2017). There, a full acceptance angle of 23 degrees for a 13 mm long chirped crystal with an IR wavelength around 1550 nm (presumably greater for 3.1 micron, and a shorter 10 mm crystal).

The mentioned work focuses on the broadband laser generation, which is different from our scope on investigating the imaging performance. This reference is included in the revised version, which supports the advantage of using chirped-poling crystals to achieve a large bandwidth.

REGARDING SINGLE-PHOTON

Mid-IR image upconversion at the single photon level is not a breakthrough either. It was already demonstrated in [21]. However, it was done in CW using only spectral filtering. Because the system is operated in pulsed mode, an additional temporal filtering is added in this work which improves S/N. However, this idea of enhancing S/N by pulse synchronization was previously reported by some of the present co-authors in [18] regarding few-photon image upconversion detection, so it can not be viewed as a new contribution of impact.

The main scope of this work is to demonstrate a large FOV while maintaining the single-photon imaging sensitivity. These superior imaging performances are made possible thanks to our previous achievement on the noise suppression techniques. Comparing to previous demonstrations with small acceptance angles, this work overcomes the FOV bottleneck in the single-photon upconversion imaging.

REGARDING TIME-OF-FLIGHT IMAGING BY PULSE COINCIDENCE

As mentioned above, some of the co-authors of this work already presented the idea of pulse coincidence for few photon detection in [18], which directly adds time-of-flight imaging capabilities. Although it is expected that the need of pulse coincidence (also used in [*]) adds time-of-flight imaging capabilities, the authors present a nice example in a supplementary video by making a transmission and reflection IR image of a target image enter the upconversion system and using time delay discrimination based on a physical change of paths.

The MIR three-dimensional imaging is demanded in tomographic characterization for highly scattering materials such as ceramics, paints, and printed electronics. Our work provides an alternation to the MIR optical coherence tomography for addressing an important need in the characterization of structured materials, thus adding additional credit to the implemented upconversion imaging system.

SCIENTIFIC QUALITY

The manuscript is scientifically correct in my opinion, although there are some minor issues that I comment next:

The 1.4 mm diameter beam is cropped to 1 mm only in the vertical direction. This flattens the intensity distribution of the pump beam in that direction (the flattening is not too strong, so the term flat-top may be questionable). The horizontal direction becomes uncropped, and has a slightly larger (1.4 times) beam diameter. It is only in the horizontal direction that resolution can be increased due to the larger soft gaussian aperture in the Fourier plane. In my opinion this should be commented in the text.

We follow the reviewer's suggestion, and clarify this point in Supplementary Note 1: *“Note that the pump size is larger than the thickness of the nonlinear crystal, which results in a cropped beam in the vertical direction.”*

Assuming an uncropped 1 mm pump beam diameter at $1/e^2$ intensity and a 1.4 mm beam cropped to 1 mm in the vertical direction, there is not a difference in the highest spatial frequency allowed through the system. Only relative amplitudes of the plane wave Fourier components change (slightly). This may represent only a slight difference (raise) in contrast for intermediate spatial frequencies only. This is discussed in Chen Yang et al., “Frequency up-conversion of an infrared image via a flat-top pump beam”, Opt. Comm. 460, 125143 (2020).

Simply stating in the text that “The cropped beam leads to a flat-top profile, thus improving the spatial resolution” is not strictly correct and requires clarification and/or referencing.

The vertical and horizontal resolutions are measured to be 125 and 99 μm , as shown in Figure R2. The plot and related discussion are added in Supplementary Note 2. In the main text, we also add the relevant reference as: *“The cropped beam within the crystal shows a larger size along the horizontal direction than the vertical one, which leads to a slightly better horizontal resolution [R6].”*

[R6] Yang, C., Liu, S.-L., Zhou, Z.-Y., Li, Y., Li, Y.-H., Liu, S.-K., Xu, Z.-H., Guo, G.-C. & Shi, B.-S.

Figure R2: (a) Recorded image for a resolution target. (b) Zoom-in for the central part. The dashed boxes in blue and yellow indicate the resolution limits along the horizontal and vertical directions.

Other issues:

If I am not wrong, the focal length F_2 is not given in the text or in the additional material.

We add this information in the text: *“The upconversion imaging system is configured in the $4f$ architecture, including two lenses with focal lengths of 50 and 100 mm.”*

In fig. 3 (k to o) it is shown that the CPLN is robust against temperature variation. This is an experimental verification of what is already outlined in ref [42]. The information in (a to j) is quite similar to that given experimentally and theoretically in [30].

These presented results are used to verify our theoretical simulation. This information is relevant but not the main achievement of this work. Hence, we put it in Supplementary Information for the better understanding to the readers.

SUMMARY

For the reasons I give, rather than a new contribution of impact, this work is in essence the combination of methods used in previous works by other authors or by some co-authors of this work. Therefore, this paper does not represent in my opinion an important advance or breakthrough to specialists in the field of significance enough for the usual standards of Nature Communications. There are some welcome experimental performance improvements in this work however, but only of an incremental character for specialists in the field.

It is worth publishing the article in another journal, as image up-conversion is a valuable rescued field from the late sixties and all information, progress, or reviews in the field, are valuable in my opinion.

We disagree with the reviewer. As recognized by the reviewer, the upconversion technique has long

be recognized useful to realize infrared imaging. To date, one of remaining challenges lies to enlarge the FOV of the upconversion imaging system, which is highly demanded to promote broader applications. Our work addresses this long-standing quest with a remarkable advance.

Notably, the concept of adiabatic conversion is for the first time adopted to facilitate the wide-field imaging. It is the original configuration that makes it possible to obtain the mid-infrared imaging features with large field of view, single-photon sensitivity and MHz-level frame rate. The achieved imaging performance is way beyond the reach of any reported instantiations in various protocols, which hence represents a significant breakthrough for the upconversion imager. We believe that the achieved advance fits well the standards of Nature Communications.

We thank again the referee for having given us the opportunity to clarify few points of our paper and to make more explicit the novelty of our work. We hope with these changes he/she might consider our paper for publication.

Reviewer #2

The authors present a paper for increasing the angular bandwidth of image upconversion when using monochromatic illumination; A paper which I read with great interest. Using a chirped PPLN structure (CPLN), the authors demonstrate that a large field of view (FOV) can be obtained (35.1 degrees) versus 3.2 degrees for a standard PPLN crystal (10 mm long). This is obtained without any moving parts contrasting, for instant, reference [35]. The large FOV is a highly attractive feature for many MIR applications and the demonstrated FOV approaches that of standard cameras which is excellent. The FOV is the main scientific accomplishment and provides a significant advancement compared to prior art. A special feature is the high temperature stability, which promotes easy implementation of the crystal assembly. Secondly, the authors demonstrate central features of their imaging approach for different applications. a) Impressive image acquisition update rate of 214 KHz b) Demonstration of single photon sensitivity and c) 3-D imaging. These demonstrations add significant value to the work serving as inspiration for new implementations and applications (although not entirely new).

We thank the reviewer for his/her positive evaluation and helpful comments on our work. In the following, we present the reply and related changes in the revised manuscript.

Technical comments:

Upconversion efficiency: The “price” for obtaining a large FOV using CPLN in the presented approach is a reduction in upconversion conversion efficiency compared to co-linear monochromatic upconversion using a PPLN with the same crystal length. A back of the envelope calculation suggests that the reduction in efficiency may well be in the order of 100 $(35.1/3.2)^2$. This is tolerable for some implementations, however important to consider for other situations. If possible,

- I suggest that the authors include a few remarks about the efficiency of using their CPLN implementation, preferably quantitative.

We thank the reviewer’s insightful suggestion. Indeed, for a given pump power at 0.6 W, the conversion efficiency was measured to be about 10% and 0.1% for the PPLN and CPLN crystals, respectively. In the revised version, we add the related discussion in Supplementary Note 2: *“The peak conversion efficiency is estimated to 0.1 % at the pump power of 0.6 W, which was about two orders of magnitude lower than that by using a PPLN crystal for a given pump intensity.”*

Set-up: The set-up includes a mode-locked laser, which mixes (DFG) with a CW ECDL to generate the MIR signal. The DFG peak efficiency, I expect, will be high due to the high peak power present in the YDFL, however the small duty cycle of the mode-locked laser will diminish average conversion efficiency to $< < 1$ %. This can be overcome by synchronous pulsed upconversion where pump pulse and mixing pulse is temporally matched, e.g. [35]

- The authors should list the pulse width of the YDFL pulses and preferably provide a few comments on the efficiency.

We follow the reviewer's suggestion and add the relevant information in Supplementary Information: *"In the temporal domain, the pulse duration of the pump τ_p is measured to be 8.5 ps as given in Supplementary Figure 2c by an auto-correlator (APE, pulseCheck)." and "The overall conversion efficiency for the MIR generation was limited by the small duty cycle of the mode-locked YDFL, which could be substantially improved by using the coincidence-pumping scheme [S1]. Additionally, the high sensitivity of the subsequently implemented upconversion imager alleviates the need for a high-power MIR illumination source."*

[S1] Huang, K., Wang, Y., Fang, J., Chen, H., Xu, M., Hao, Q., Yan, M. & Zeng, H. Highly efficient difference-frequency generation for mid-infrared pulses by passively synchronous seeding. High Power Laser Sci. Eng. 9, e4 (2021).

The authors nicely demonstrate the possibility to temporal filter noise using the short pulse as a time window for noise elimination. The idea is not new, but illustrate the single photon capability.

Indeed, the temporal filtering plays an important role to realize the single-photon capability. The achieved imaging performances are made possible thanks to our previous achievement on the noise suppression techniques. Comparing to previous demonstrations with small acceptance angles, this work overcomes the FOV bottleneck in the single-photon upconversion imaging.

Aberrations: Using a chirped CPLN to upconvert with a large FOV imply that different angles will be upconverted in different sections of the CPLN crystal. This is like a "thick" filter.

- Will this give rise to image aberrations? The authors should include a few comments on this issue.

In the revised manuscript, we add the related discussion in Supplementary Note 2: *"During the upconversion imaging process, the components at different incidence angles are upconverted in different sections of the CPLN crystal. Particularly, the involved poling periods for the angles from 0 to 15 degrees correspond to the range between 19.5 and 21 μm , in part of the full range from 19 to 24 μm . In the experiment, two injection directions into the CPLN crystal have been tested for comparison, which exhibit similar imaging resolution and show no associated image aberration. One possible reason may lie in the homogeneous pump field along the crystal due to the large beam size. Further investigation could be conducted by numerally solving the three-dimensional coupled-wave equations."*

Conclusion: The paper is clearly written and easy to understand, figures and graphs are of high quality. When considering the paper in its totality, it merits publication in Nature Comm. after minor revision of different points as listed above.

We thank the reviewer for his/her positive report and for recommending publication in Nature Communications.

REVIEWERS' COMMENTS

Reviewer #1 (Remarks to the Author):

The authors provide an experimental verification of a large FOV using a chirped PPLN crystal, which was otherwise predicted and estimated regarding adiabatic SFM by other authors included in the references, as pointed out in my former review. Although they do not exploit the potential of high QE (quantum efficiency) conversion in adiabatic SFM, they get enough sensitivity for few-photon detection at quite high frame rates. This is however, compatible with a future improvement of their conversion efficiency by increasing their peak power in the pump beam to achieve adiabatic SFM conditions, thus fully demonstrating the capability of a large FOV and a high QE at the same time.

When replying to my comments in my former review, the authors point out that the references I mentioned are focussed in spectral bandwidth improvement by chirped PPLN. But it is known in general to the scientific community that spectral bandwidth and QPM angle acceptances come together (surely the authors are well aware of it). In particular, in ref [42] not only a spectral bandwidth is reported as the main point, but also an associated FOV of 25 degrees (or more) for a monochromatic illumination is revealed based on simulation (Section 3, page 12739, which is essentially the same FOV value the authors theoretically/experimentally report.

Although I still think that the main contribution is an experimental verification of various things reported previously, I believe that the image up-conversion field is very useful, and that any additional information is welcome in the field. Thus, I do not oppose to publication of the manuscript in Nature Communications in its present form, and leave the decision to the editor.

In my opinion the manuscript is of high quality, well organized and presented, and the multimedia material provided is excellent.

Reviewer #2 (Remarks to the Author):

Questions/comments have been addressed. I suggest submission as is.

Manuscript NCOMMS-21-34014A
“Wide-field mid-infrared single-photon upconversion imaging”
Reply to the Reviewers

Reviewer #1

The authors provide an experimental verification of a large FOV using a chirped PPLN crystal, which was otherwise predicted and estimated regarding adiabatic SFM by other authors included in the references, as pointed out in my former review. Although they do not exploit the potential of high QE (quantum efficiency) conversion in adiabatic SFM, they get enough sensitivity for few-photon detection at quite high frame rates. This is however, compatible with a future improvement of their conversion efficiency by increasing their peak power in the pump beam to achieve adiabatic SFM conditions, thus fully demonstrating the capability of a large FOV and a high QE at the same time.

Our work addresses the long-standing quest for the infrared upconversion imaging to realize a large field of view while maintaining a high imaging sensitivity. Notably, the concept of adiabatic conversion is adopted for the first time to facilitate the wide-field imaging, which is beyond the scope of previous demonstrations on the broadband spectral conversion. As the reviewer pointed out, the proposed scheme holds the potential to implement a wide-field infrared imaging with a high conversion efficiency by further augmenting the pump power.

When replying to my comments in my former review, the authors point out that the references I mentioned are focused in spectral bandwidth improvement by chirped PPLN. But it is known in general to the scientific community that spectral bandwidth and QPM angle acceptances come together (surely the authors are well aware of it). In particular, in ref [42] not only a spectral bandwidth is reported as the main point, but also an associated FOV of 25 degrees (or more) for a monochromatic illumination is revealed based on simulation (Section 3, page 12739, which is essentially the same FOV value the authors theoretically/experimentally report.

Ref. [42] theoretically verified the improved tolerance for the incidence angle, with an aim to highlight the robustness of the adiabatic conversion. In contrast, our work presents theoretical and experimental investigation on the wide-field imaging performance.

Although I still think that the main contribution is an experimental verification of various things reported previously, I believe that the image up-conversion field is very useful, and that any additional information is welcome in the field. Thus, I do not oppose to publication of the manuscript in Nature Communications in its present form, and leave the decision to the editor.

In my opinion the manuscript is of high quality, well organized and presented, and the multimedia material provided is excellent.

Our work unprecedentedly explores the possibility of the adiabatic conversion in facilitating a wide-

field upconversion imaging. The achieved imaging features thus represent a significant breakthrough for the upconversion imager. We thank the reviewer for his/her positive evaluation on our presented results. In the present form of manuscript, we have carefully revised to elaborate the novelty of our work. With these improvements, we hope that our work could finally be accepted for publication in Nature Communications.

Reviewer #2

Questions/comments have been addressed. I suggest submission as is.

We thank again the reviewer for recommending publication in Nature Communications.